# Gut Microbiota in Chronic Kidney Disease: From Composition to Modulation towards Better Outcomes—A Systematic Review

**DOI:** 10.3390/jcm12051948

**Published:** 2023-03-01

**Authors:** Luminita Voroneanu, Alexandru Burlacu, Crischentian Brinza, Andreea Covic, Gheorghe G. Balan, Ionut Nistor, Cristina Popa, Simona Hogas, Adrian Covic

**Affiliations:** 1Nephrology Department, Dialysis and Renal Transplant Center, “Dr. C.I. Parhon” University Hospital, 700503 Iasi, Romania; 2Faculty of Medicine, ‘Grigore T. Popa’ University of Medicine, 700115 Iasi, Romania; 3Department of Interventional Cardiology, Cardiovascular Diseases Institute “Prof. Dr. George I.M. Georgescu”, 700503 Iasi, Romania; 4Institute of Gastroenterology and Hepatology, St. 1 Spiridon Emergency County Hospital, 700111 Iasi, Romania

**Keywords:** chronic kidney disease, hemodialysis, end-stage kidney disease, gut microbiota, gut dysbiosis, mortality, outcomes

## Abstract

Background: A bidirectional kidney–gut axis was described in patients with chronic kidney disease (CKD). On the one hand, gut dysbiosis could promote CKD progression, but on the other hand, studies reported specific gut microbiota alterations linked to CKD. Therefore, we aimed to systematically review the literature on gut microbiota composition in CKD patients, including those with advanced CKD stages and end-stage kidney disease (ESKD), possibilities to shift gut microbiota, and its impact on clinical outcomes. Materials and methods: We performed a literature search in MEDLINE, Embase, Scopus, and Cochrane databases to find eligible studies using pre-specified keywords. Additionally, key inclusion and exclusion criteria were pre-defined to guide the eligibility assessment. Results: We retrieved 69 eligible studies which met all inclusion criteria and were analyzed in the present systematic review. Microbiota diversity was decreased in CKD patients as compared to healthy individuals. Ruminococcus and Roseburia had good power to discriminate between CKD patients and healthy controls (AUC = 0.771 and AUC = 0.803, respectively). Roseburia abundance was consistently decreased in CKD patients, especially in those with ESKD (*p* < 0.001). A model based on 25 microbiota dissimilarities had an excellent predictive power for diabetic nephropathy (AUC = 0.972). Several microbiota patterns were observed in deceased ESKD patients as compared to the survivor group (increased Lactobacillus, Yersinia, and decreased Bacteroides and Phascolarctobacterium levels). Additionally, gut dysbiosis was associated with peritonitis and enhanced inflammatory activity. In addition, some studies documented a beneficial effect on gut flora composition attributed to synbiotic and probiotic therapies. Large randomized clinical trials are required to investigate the impact of different microbiota modulation strategies on gut microflora composition and subsequent clinical outcomes. Conclusions: Patients with CKD had an altered gut microbiome profile, even at early disease stages. Different abundance at genera and species levels could be used in clinical models to discriminate between healthy individuals and patients with CKD. ESKD patients with an increased mortality risk could be identified through gut microbiota analysis. Modulation therapy studies are warranted.

## 1. Introduction

Gut microbiota represents one of the most diverse microbiota of the human body and encompasses more than 35,000 bacterial species with 10 million genes [1]. For that reason, gut microbiota has been referred to by some authors as an additional organ and has been extensively studied in recent years [2,3,4].

Although gut microbiota varies across individuals, the most frequently encountered phyla are *Firmicutes* and *Bacteroidetes*, which constitute approximately 90% of the microbiota. Other gut microbiota phyla are represented by *Actinobacteria*, *Fusobacteria*, *Proteobacteria*, and *Verrucomicrobia* [5]. Among the *Firmicutes* phylum, *Clostridium*, *Lactobacillus*, *Bacillus*, *Ruminococcus*, and *Enterococcus* are the most frequent genera.

In addition to the local effects attributed to gut microbiota, it could also have systemic effects through secreting different active compounds, including short-chain fatty acids (SCFA) (acetate, butyrate, propionate), neurotransmitters (dopamine, serotonin, noradrenaline), bile acids, trimethylamine, cortisol, and gastrointestinal hormones (glucagon-like peptide-1, leptin, peptide YY) [4]. Therefore, gut microbiota could be regarded as a genuine endocrine organ that modulates nutrient and drug metabolism, antimicrobial protection, and immune response and ensures the integrity of the gastrointestinal tract [1,4].

In CKD patients, a bidirectional kidney–gut axis has been described [6]. The underlying cause for renal dysfunction, dietary restrictions, prolonged colonic transition time, or therapeutic intervention such as antibiotics, iron supplementation, or phosphate binders could cause dysbiosis.

Alternatively, gut dysbiosis triggers the production of detrimental metabolites such as indoxyl sulfate (IS) and p-cresyl sulfate, already associated with increased mortality and cardiovascular risk and a reduced number of valuable SCFA; the latter is implicated in energy homeostasis, maintaining the gut barrier, blood pressure control, and immune regulation. Moreover, dysbiosis induces an increase in gut permeability, which favors the translocation of bacterial species and microbial products through systemic circulation, promoting systemic inflammation and, possibly, alterations of glucose and lipid metabolism [7]. This leaky gut syndrome has also been observed in patients with inflammatory bowel disease, colorectal cancer, Parkinson’s disease, and Huntington’s disease [8,9]. An additional unfavorable effect of dysbiosis is the loss of diversity and imbalance in composition. This was related to poor survival in patients undergoing allogeneic hematopoietic-cell transplantation [10] and in patients hospitalized for chronic obstructive pulmonary disease [11].

Moreover, a heart–gut axis was described, linked to atherosclerosis and heart failure pathogenesis and development [12]. Gut dysbiosis observed in patients with chronic kidney disease (CKD) might partly explain the increased rate of cardiovascular-related deaths (almost 35% of all deaths) in this subgroup of patients [13,14]. Thus, gut dysbiosis could be regarded as a potential cardiovascular risk factor in patients with CKD in addition to other traditional risk factors. Therefore, the interplay between the gut, kidney, and heart is of great clinical importance, as gut microbiota composition could be modulated by diet, physical activity, probiotics, and prebiotics [12].

Consequently, we aimed to systematically review the literature on gut microbiota composition in CKD patients, including those with advanced CKD stages and end-stage kidney disease (ESKD), possibilities to shift gut microbiota, and its impact on clinical outcomes.

## 2. Materials and Methods

We conducted our systematic review in line with the updated Preferred Reporting Items for Systematic Review and Meta-analysis (PRISMA) guidelines in order to obtain reliable results [15]. The protocol of the systematic review was registered in the PROSPERO database (CRD42022369573).

### 2.1. Data Sources and Search Strategy

We sought to find eligible studies in MEDLINE (PubMed), Embase, Scopus, and Cochrane databases. A literature search was performed in the aforementioned databases, using various combinations between pre-specified keywords and MeSH terms or Emtree terms (respectively, for MEDLINE or Embase database): “microbiota”, “gut”, “microbiome”, “gastrointestinal”, “microflora”, “intestinal”, “composition”, “diversity”, “chronic kidney disease”, “end-stage kidney disease”, “hemodialysis”, “renal decline”, “disease severity”, “progression”, “mortality”, “major adverse cardiovascular events”, “inflammation”, “inflammatory response”, “probiotic”, “prebiotic”, and “symbiotic”. Studies were published from the inception of databases to 30 June 2022. We did not apply any language filters or restrictions in the search strategy. In addition, the ClinicalTrials.gov registry of clinical trials, Google Scholar engine, and references from cited manuscripts were screened and checked for additional eligible studies, which is compliant with PRISMA guidelines. The final search strategy for all databases and references obtained is presented in Appendix A.

### 2.2. Eligibility Criteria and Outcomes

A multistep approach was used to assess retrieved references for eligibility. In the first step, two independent investigators evaluated the title and abstracts of articles for inclusion and exclusion criteria. In the next step, the full text of studies that met the eligibility criteria based on title and abstract was appraised.

Several inclusion criteria were pre-defined and were applied for eligibility assessment. (1) Both randomized clinical trials and observational studies were considered for inclusion. (2) Patients ≥ 18 years old with CKD of all stages were enrolled. (3) Healthy subjects or patients with early stages of CKD were included in the control group (when available). (4) Studies reporting original data on the following outcomes: (a) gut microbiota composition in CKD patients (when available, CKD patients versus healthy subjects or early CKD patients versus ESKD patients); (b) association between identified microbiota species in CKD patients and major adverse cardiovascular events (MACE), mortality, CKD severity, and disease progression; (c) the impact of prebiotics, probiotics, and symbiotics on flora composition and outcomes of CKD patients.

Early CKD patients were considered those presenting with G2 and G3 CKD stages (estimated glomerular filtration rate of 30–90 mL/min/1.73 m^2^). Prebiotics were defined as “non-digestible food ingredient” which enhance the growth or activity of certain beneficial gut bacteria [16]. Probiotics were defined as “live microorganisms” which exhibit beneficial health effects when they are prescribed in appropriate concentration [17]. Likewise, we defined synbiotics as a “mixture of probiotics and prebiotics” which display beneficial health effects [18].

In addition, some key exclusion criteria were established: unpublished data, studies available only in abstract, overlapping populations, case reports, meta-analyses, editorials, missing data, and inability to extract data regarding the population enrolled and outcomes investigated.

### 2.3. Data Collection and Synthesis

After eligibility assessment, two independent investigators extracted the following data from included studies: first author, year of publication, study design, number of patients enrolled and their age, clinical setting, reported outcome of interest, and follow-up period. We performed a qualitative synthesis of included studies to provide a better understanding of reported outcomes. Additionally, when available, data were reported as numbers, median and mean values, odds ratio (OR), and *p*-value.

### 2.4. Quality and Risk of Bias Assessment

In the case of randomized controlled clinical trials, the risk of bias was judged using the revised Cochrane risk-of-bias tool for randomized trials (RoB 2) [19]. The Newcastle–Ottawa scale (NOS) was used to guide the quality assessment of observational, non-randomized studies. NOS is a tool based on designating stars for signaling questions, which were grouped into three domains: population sampling, comparability of groups, and evaluation of outcomes of interest [20].

## 3. Results

We searched the specified databases and retrieved 3290 references. Duplicate records were removed (*n* = 1927), leaving 1363 references for title or abstract screening. Finally, 121 records were assessed for eligibility in full-text, and 69 studies were included in the present systematic review. The flowchart of the selection process is presented in Figure 1.

General data of the analyzed studies, including publication year, study design, number of patients enrolled and their age, as well as gut microbiota composition in CKD patients, are reported in Table 1. The majority of included studies had an observational non-randomized design [21,22,23,24,25,26,27,28,29,30,31,32,33,34,35,36,37,38,39,40,41,42,43,44,45,46,47,48,49,50,51,52,53,54,55,56,57,58,59,60,61,62,63,64,65,66,67,68,69,70,71,72,73,74,75,76,77], while 12 studies were randomized trials [78,79,80,81,82,83,84,85,86,87,88,89]. Additionally, 21 studies investigated gut microbiota composition in ESKD patients (including patients with hemodialysis or peritoneal dialysis) [24,28,29,30,31,33,34,40,41,42,45,52,53,54,55,60,62,78,83]. Moreover, 4 studies investigated gut microbiota differences in early CKD stages as compared to healthy controls [30,51,56] or advanced-stage CKD [38].

### 3.1. Early-Stage CKD

Hu et al. observed different gut microbiota compositions even in patients with early-stage CKD as compared to healthy controls [74]. Gut flora diversity was significantly decreased in early-stage CKD patients (*p* < 0.001) compared to those without CKD. At the genera level, *Ruminococcus* had a good power to discriminate between early-stage CKD patients and healthy controls (AUC = 0.771, 95% CI, 0.771–0.852), while *Roseburia* accurately identified healthy controls (AUC = 0.803, 95% CI, 0.804–0.864) [74]. Wu et al. reported similar discriminatory capacity for early-stage CKD patients in the case of *Bacteroides eggerthii* (AUC = 0.80, 95% CI, 0.67–0.93), which was higher than in the case of protein/creatinine ratio (AUC = 0.64) [56]. Studies investigating gut microbiota diversity are of particular importance, as decreased diversity could be considered a reliable marker of gut dysbiosis [90]. Consequently, even patients with early-stage CKD had gut dysbiosis (low microbiota diversity) when compared to healthy controls [74]. Thus, improving gut microbiota diversity constitutes an important target for future interventional studies involving CKD patients. Available data on *Roseburia* suggest a key role in the modulation of gut barrier homeostasis and inflammation [91]. Hence, *Roseburia* is a marker of a normal gut microbiome, and lower abundance was reported in CKD patients from early stages [74]. Moreover, *Roseburia* abundance decreases along with CKD progression (lower in ESKD patients compared to early-stage CKD) [53].

### 3.2. ESKD

Some studies reported a shift in gut microbiota profile along with CKD progression, especially in ESKD patients and those requiring renal replacement therapy (RRT). Gao et al. reported a progressively increasing abundance of *Bifidobacterium* in advanced CKD stages, while *Lactobacillus* levels decreased [24]. He et al. observed a lower abundance of both *Bifidobacterium* and *Lactobacillus* in ESKD patients compared to healthy controls. Nevertheless, *Bifidobacterium* and *Lactobacillus* were increased in hemodialysis patients compared to ESKD patients without RRT [29]. Most studies documented decreased levels of *Roseburia* in advanced CKD stages, including ESKD [33,52,53,54,55,60,62]. Additionally, *Roseburia* abundance was lower in ESKD patients as compared to those with CKD stages 1–4 and healthy controls (*p* < 0.001) [53]. *Bifidobacterium*, like other beneficial components of a normal gut microbiome, exerts modulatory effects on gut homeostasis, inflammation, and immune response [92]. Although some studies documented *Bifidobacterium* depletion in CKD patients (including ESKD), these results should be confirmed in larger clinical trials.

### 3.3. Diabetic Nephropathy

Patients with diabetic nephropathy stage 3–4 had a different microbiota profile compared to healthy individuals [23]. Du et al. developed a model based on 25 gut microbiota dissimilarities, which had an excellent predictive power for diabetic nephropathy (AUC = 0.972) [23]. Additionally, in patients with diabetes-associated CKD, *Ruminococcaceae* and *Bacteroidaceae* abundance was significantly increased, while *Prevotellaceae* levels were decreased. These microbiome alterations were observed across all CKD stages, highlighting early gut dysbiosis in diabetic patients, which maintains in advanced CKD stages [38]. In addition, hemodialysis patients with diabetes had an increased abundance of *Desulfovibrionaceae*, *Veillonellaceae*, and *Lactobacillaceae* as compared to hemodialysis patients without diabetes [83].

Outcomes of CKD patients (inflammation, renal function, disease progression, mortality, and peritonitis) linked to gut microbiota composition are displayed in Table 2. The Simpson index and the Shannon index were significantly lower in deceased hemodialysis patients, as compared to those who survived (respectively, *p* = 0.007 and *p* = 0.028). Moreover, several bacteria were increased in hemodialysis patients from the non-survivor group (*Oscillospira*, *Achromobacter*, *Agrobacterium*, *Lactobacillus*, *Alloscardovia*, *Anoxybacillus*, *Devosia*, *Yersinia*) [40]. Additionally, Luo et al. reported a lower abundance of *Bacteroides* and *Phascolarctobacterium* in ESKD patients with cardiovascular mortality compared to those who survived (*p* < 0.05 for both).

Peritonitis was associated with altered gut microbiota composition in peritoneal dialysis patients, as was reported by some authors. Zhou et al. observed a higher abundance of *Bacteroidetes* and *Synergistetes* in the peritonitis group as compared to patients without peritonitis, while *Bacilli* and *Lactobacillus* were decreased [70]. Additionally, *Dorea* and *Clostridium* abundance was decreased in peritoneal dialysis patients [45].

Moreover, available gut microbiota modulation interventions (including dietary interventions, probiotics, prebiotics, and synbiotics) are presented in Table 3. Concerning intestinal flora modulation, synbiotics increased *Bifidobacterium* levels up to 5-fold from baseline (*p* = 0.003) [87], which was concordant in clinical studies [84,85]. *Lactobacillus* levels were decreased following synbiotic therapy in one study [85], whereas in another study, *Lactobacillus* abundance was similar before and after the treatment [87]. Nevertheless, synbiotic therapy was linked to eGFR decrease with 3.14 mL/min/1.73 m^2^ (*p* < 0.01), requiring further research [84].

Liu et al. observed that probiotics increased *Bacteroidaceae* and *Enterococcaceae* abundance in hemodialysis patients, while *Ruminococcaceae*, *Halomonadaceae*, *Peptostreptococcaceae*, and *Clostridiales* family were decreased [83]. Additionally, probiotics significantly increased *Lactobacillales* and *Bifidobacteria* levels (*p* < 0.001) in another study [88].

The quality of non-randomized studies was fair to good as assessed by NOS adapted for cross-sectional, case-control, and cohort studies (Appendix A). The risk of bias in randomized trials was appraised using the RoB 2 tool and is displayed in Figure 2.

## 4. Discussion

This study assessed gut microbiota alteration across a spectrum of kidney disease through a systematic review. The main findings are: (1) a different microbiota composition and a decreased gut diversity were reported from early stages to advanced CKD; (2) patients with CKD shared the depletion of anti-inflammatory butyrate-producing microbes (i.e., *Roseburia*, *Prevotella*, *Bacteroides*) and the enrichment of pro-inflammatory microbes (*Proteobacteria*, *Actinobacteria*); (3) there are limited data regarding the impact of dysbiosis on inflammation, mortality, or cardiovascular risk.

This review was designed to comprehensively illustrate the alteration of the gut microbiome in CKD patients. In more than half of the analyzed studies, gut diversity was significantly decreased. Similar data were reported last year by Zhao et al. [93]. Six (6/9) and four studies (4/6) that included patients with CKD and ESKD, respectively, suggested that the α-diversity of gut microbiota was significantly lower in patients than in healthy controls. Additionally, ten (10/11) studies that analyzed patients with CKD and five (5/6) studies that focused on ESKD reported a significantly altered composition of the gut microbiota in patients as compared to healthy controls [93].

At the phylum level, *Fusobacteria*, *Verrucomicrobia*, and *Proteobacteria* abundances were significantly higher in CKD [39,61]. At the genus and species levels, there were some substantial differences; probably, the individual variances determined by genetic and environmental factors and the etiology and the severity of CKD could explain the variation and inconsistency of the studies.

*Proteobacteria*, including common bacteria such as *Escherichia coli*, *Salmonella*, or *Desulfovibrio*, are increased in CKD [49,83]. An increased level was associated with abnormal gut barrier function, which might result in increased epithelial permeability, allowing microbial fragments and products to enter the sub-epithelial space and lamina propria, lipopolysaccharides translocation, and enhancing inflammatory response [94]. Additionally, people with an increased level of *Desulfovibrio* spp. had more severe renal dysfunction [94].

Concerning phylum *Firmicutes*, we noticed an increase in the genus *Streptococcus* and a lower abundance of the genus *Roseburia*, *Faecalibacterium*, and *Prevotella*. *Roseburia* produces butyrate, which could promote the proliferation of extrathymic regulatory T cells (Tregs). Tregs, as vital anti-inflammatory lymphocytes, produce interleukin-10, transforming growth factor beta, and interferon gamma. Microbial butyrate has been established to contribute to the pro- and anti-inflammatory balance by inducing Tregs differentiation [95]. *Roseburia* was negatively related to inflammatory status, renal function, and CKD progression [32,33].

From the phylum *Verrucomicrobia*, the genus *Akkermansia* plays an essential role in improving gut-barrier function and viscosity of the mucus. It also promotes the growth of bacteria-producing SCFAs, such as butyrate, by offering them carbon, nitrogen, and energy produced as a consequence of mucus degradation [96]. It also has anti-inflammatory properties. Unfortunately, the abundance was decreased in CKD patients [41,83] and was negatively related to inflammation [46].

Regarding the *Bacteroidetes* phylum, we observed an increase in the genus *Bacteroides* and a lower abundance of the genus *Prevotella*. In the general population, it was noticed that the Western diet was associated with *Bacteroides* and *Clostridiales* abundance in the gut microbiome, while rural populations with a high-fiber, low-protein diet tended to have *Prevotella*, which can produce SCFA. The authors from one study evaluated fecal microbiota composition differences between ESKD patients and 60 healthy controls. They found that *Prevotella* was enriched in the healthy group, whereas *Bacteroides* was prevalent in the ESKD group. Moreover, *Prevotella* was negatively related to inflammatory status and renal function [33]. *Bacteroides* was increased in most studies, whereas only two studies reported decreased levels [48,53]. Further analysis confirmed that *Bacteroides* was related to cardiovascular mortality in patients with dialysis. Nevertheless, the authors reported only five cardiovascular-related deaths, thus limiting the extrapolation of the results to all CKD patients. Consequently, these data should be confirmed in large clinical trials [45]. In addition, patients with peritoneal dialysis and peritonitis had a higher abundance of *Bacteroides* compared with the non-peritonitis group [70].

Our data are comparable to those obtained by Zhao et al.; the same abundance of Proteobacteria in CKD and ESKD was identified [93]. However, we identified a greater abundance of *Bacteroides* in CKD patients as compared to previous data. Zhao et al. reported enriched levels of bacteria from the Bacteroides genus in 3/11 studies involving CKD patients and in 3/9 studies on ESKD patients [93]. Furthermore, it was connected with infections and cardiovascular mortality [93].

Gut microbial dysbiosis has also been reported in diabetes mellitus. Patients with diabetic nephropathy exhibited increased levels of multiple pathogenic genera such as *Acidaminococcus*, *Lactobacillus*, *Megasphaera*, *Clostridium*, *Sutterella*, and *Desulfovibrionaceae*, while healthy controls had a high abundance of butyrate-producing bacteria [23]. The data are similar to those from a recently published systematic review [97]. Nevertheless, additional studies are warranted to investigate which specific microbes are involved in the pathophysiology of CKD linked to diabetes mellitus.

Data concerning phosphate binders on the gut microbiome in CKD patients are limited. Two studies documented a lower gut flora diversity in HD patients receiving phosphate binder [63,65]. In another study, sucroferric oxyhydroxide supplementation increased levels of *Ruminococcus torques*, which could influence gut barrier permeability [75,98]. Likewise, after oral iron supplementation, α-diversity and *Firmicutes* levels decreased, while *Bacteroides* increased [76]. These findings warrant further safety analysis of phosphate binders and iron supplementation in terms of gut homeostasis and microbiome composition in CKD patients.

Evidence sustaining the benefit of probiotic, prebiotic, and synbiotic supplementation in the management of CKD is mixed. Some studies insinuated that they might be useful by decreasing uremic and inflammatory toxins [99]. They could also improve oxidative stress, as well as lipid profiles in patients with CKD, which are well-known cardiovascular risk factors [100].

## 5. Conclusions

In conclusion, patients with CKD had an altered gut microbiome profile, even at early disease stages, as was documented consistently in clinical studies. Moreover, studies reported a shift in gut microbiota composition along with CKD progression, especially in ESKD patients and those requiring RRT. Different abundance at genera and species levels could be used in clinical models to discriminate between healthy individuals and patients with CKD (including those with diabetic nephropathy), with excellent predictive power. In addition to traditional risk factors, ESKD patients with an increased mortality risk could also be identified through gut microbiota analysis. In addition, studies established gut microbiome patterns linked to enhanced inflammatory activity and to a higher risk of peritonitis in patients receiving peritoneal dialysis. Nevertheless, clinical studies with larger sample sizes are required to confirm the association between altered gut microbiota composition and adverse outcomes in CKD patients, including all-cause and cardiovascular mortality. Due to data inconsistency, randomized clinical trials are needed to analyze the effect of different microbiota modulation therapies on gut bacterial composition and adverse end-points in CKD patients.

## Figures and Tables

**Figure 1 jcm-12-01948-f001:**
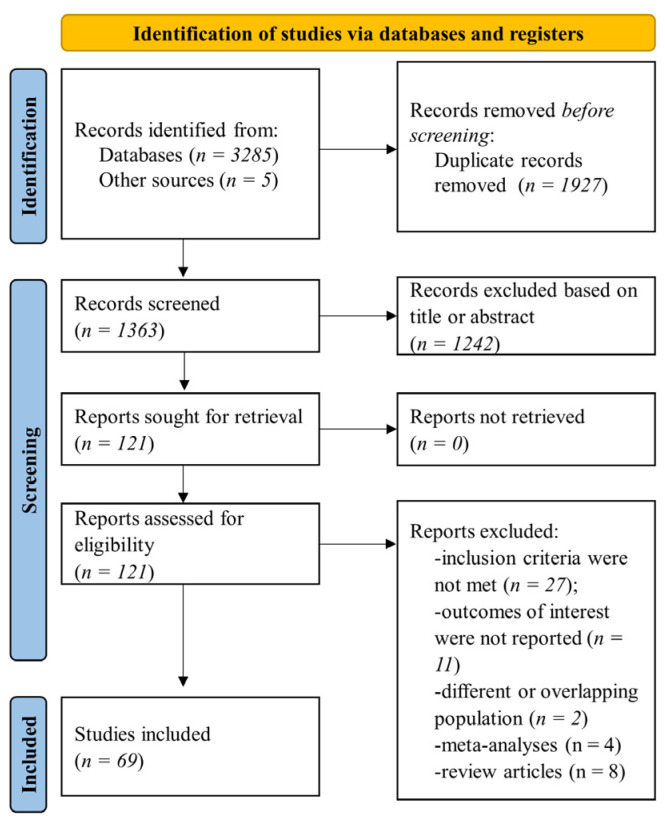
Flow diagram of selected studies in the present analysis.

**Figure 2 jcm-12-01948-f002:**
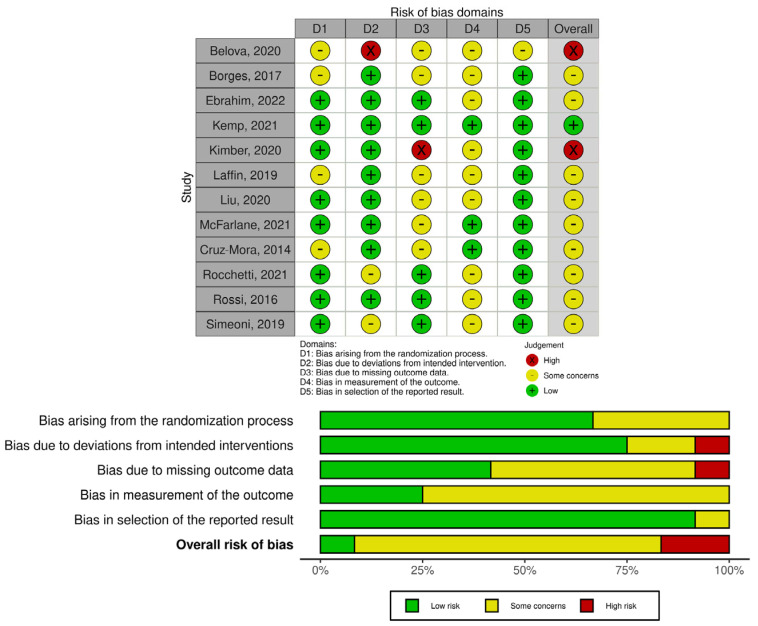
Risk of bias of randomized trials (RoB 2). Belova et. al., 2020 [78], Borges et al., 2017 [79], Ebrahim et al., 2022 [80], Kemp et al., 2021 [81], Kimber et al., 2020 [82], Laffin et al., 2019 [89], Liu et al., 2020 [83], McFarlane et al., 2021 [84], Cruz-Mora et al., 2014 [85], Rocchetti et al., 2021 [86], Rossi et al., 2016 [87], Simeoni et al., 2019 [88].

**Table 1 jcm-12-01948-t001:** Gut microbiota composition in CKD patients as reported in analyzed studies.

Author, Year	Design	Patients, No	Age, Median/Mean ± SD	Gut Microbiota Composition
Barros et al., 2015 [21]	Cross-sectional study	20 (CKD stage 3–4)	64.4 ± 9.1	Similar number of bands in patients with CKD vs. healthy controls. CKD: *Listeria monocytogenes*, *Flavobacteriaceae bacterium*. Healthy controls: *Uncultured Lachnospiraceae bacterium*, *Butyrivibrio crossotus*.
		19 (healthy participants)	51.6 ± 6.6	
Belova et al., 2020 [78]	Randomized, parallel-group, controlled trial	32 (HD patients who received basic therapy + symbiotic)	57.1 ± 7.9	HD patients displayed an increased number and diversity of *Bacteroides* spp., *Clostridium* spp., *Collinsella* spp., *Eggerthella* spp., and other bacteria. Grade I dysbiosis was observed in 37.5% of patients, grade II dysbiosis in 50.0% of patients, and grade III dysbiosis in 12.5% of patients.
		30 (HD patients who received basic therapy + placebo)	54.7 ± 8.4	
Chen et al., 2021 [22]	Cross-sectional study	96 (CKD stage 1–5)	71 (CKD stage 3–5)	No significant difference was observed in α-diversity across different groups. Relative levels of *Streptococcus*, *Klebsiella pneumonia*, and *Haemophilus* *Parainfluenzae* increased progressively in advanced CKD. Increased levels of *Fusobacterium varium* and *Fusobacterium mortiferum* were observed in CKD stages 3–5.
		60 (healthy participants)	66	
Du et al., 2021 [23]	Cross-sectional study	43 (diabetic nephropathy stage 3–4)	60.86 ± 5.69	In patients with diabetic nephropathy, several genera were more abundant as compared to healthy individuals: *Acidaminococcus*, *Lactobacillus*, *Megasphaera*, *Mitsuokella*, *Olsenella*, *Prevotella_7*, and *Sutterella*. A model based on 25 genera discrepancies had a good prediction power for diabetic nephropathy (AUC = 0.972).
		37 (healthy individuals)	61.78 ± 6.40	
Ebrahim et al., 2022 [80]	Randomized controlled trial	59 (CKD stage 3–5)	41.0 ± 11.6	In CKD patients, several genera were abundant: *Faecalibacterium*, *Prevotella*, *Bacteroides*, *Blautia*, and *Roseburia*.
Gao et al., 2021 [24]	Cross-sectional study	52 (CKD stage 3–5, including 10 patients with ESKD)	–	*Eubacterium rectale* and *Collinsealla* genera correlated with kidney disease severity. *Bifidobacterium* increased with kidney disease severity, while *Lactobacillus* decreased. *Methanobacteria* was abundant in advanced CKD stages but was not present in ESKD patients.
Gryp et al., 2021 [25]	Cross-sectional study	111 (CKD stage 1–5)	–	*Faecalibacterium*, *Bacteroides*, and *Roseburia* were the most abundant genera in all CKD groups. *Butyricicoccus* (butyrate-generating properties) decreased in CKD stage 4–5 as compared to CKD stage 1–2 (*p* = 0.043).
Gryp et al., 2020 [26]	Cross-sectional study	138 (CKD stage 1–5) and 14 controls	–	*A. muciniphila*, *C. dicile*, *Enterobacteriaceae*, *Lactobacillus* spp., and *Streptococcus* spp. were increased in HD patients compared to other CKD stages. *Bifidobacterium* spp. and *Streptococcus* spp. decreased with kidney function decline, while *Enterobacteriaceae* and *E. coli* increased.
Guirong et al., 2018 [27]	Cross-sectional study	16 (kidney transplant)	42.8 ± 11.5	Microbial richness was lowest in kidney transplant patients (Chao1 index 249.6 ± 118.7) as compared to CKD group (Chao1 index 286.4 ± 89.3) and healthy controls (Chao1 index 394.5 ± 86.8). *Bacteroides* and *Enterobacteriaceae* abundance were increased in CKD and kidney transplant patients, while *Lachnospira*, *Ruminococcaceae*, and *Faecalibacterium* levels were decreased. Gut microbiota profile had a good power to discriminate between CKD and healthy controls (AUC 0.921).
		84 (CKD)	55.9 ± 18.2	
		53 (healthy controls)	54.7 ± 12.8	
Hanifi et al., 2021 [28]	Cross-sectional study	20 (CKD or ESKD)	53.20 ± 12.03	CKD and ESKD patients had similar abundance of different *Bifidobacteriaceae* species as compared to those without CKD or ESKD.
		20 (non-CKD/ESKD)	59.3 ± 7.89	
He et al., 2020 [73]	Cross-sectional study	109 (ESKD patients)	56.8 ± 15.5 (HD patients)	*Bifidobacterium* and *Lactobacillus* acidophilus decreased in ESKD patients as compared to healthy controls, while *Escherichia coli* and *Enterococcus faecalis* increased. *Bifidobacterium* and *Lactobacillus acidophilus* were higher in HD patients as compared to ESKD patients without RRT, while *Escherichia coli* and *Enterococcus faecalis* were decreased.
He et al., 2021 [29]	Cross-sectional study	30 (HD)	56.3 ± 13.6	Patients with CKD (including HD) had a higher abundance of *Escherichia coli* and *Enterococcus faecalis* compared to healthy controls (*p* < 0.05), while *Bifidobacterium* and *Lactobacillus acidophilus* were decreased (*p* < 0.05).
		24 (non-HD)	57.2 ± 15.1	
		30 (healthy controls)	57.4 ± 14.9	
Hu et al., 2020 [71]	Cross-sectional study	95 (CKD stages 1–5, non-HD)	57.45 ± 11.68 (CKD stage 5)	Several genera (*Escherichia-Shigella*, *Parabacteroides*, *Roseburia*, *Pyramidobacter rectale_group*, *Ruminococcaceae_NK4A214_group*, *Prevotellaceae_UCG.001*, *Hungatella*, *Intestinimonas*, *Pyramidobacter*) discriminated between CKD stage 5 and healthy controls (AUC = 0.938, 95% CI, 0.853–1.000). Patients with CKD exhibited increased levels of *Proteobacteria* and decreased levels of *Synergistetes* as compared to healthy controls.
Hu et al., 2020 [74]	Case-control study	47 (early CKD)	43.2 ± 12.6	α-diversity was lower in CKD versus healthy control participants. *Proteobacteria* and *Actinobacteria* were increased in CKD group vs. control group. Thirty-one species were different in CKD patients compared to healthy control (highest diagnostic power for *Ruminococcus* and *Roseburia*). *Ruminococcus* displayed the highest AUC for CKD prediction (0.771, 95% CI, 0.771–0.852).
		150 (healthy controls)	38.5 ± 15.4	
Hu et al., 2020 [30]	Case-control study	166 (47 non-dialysis CKD, 49 HD group, 53 PD group, and 17 healthy controls)	57.80 ± 10.03	α-diversity and β-diversity were lower in PD patients as compared to HD patients and control participants. *Enterobacteriaceae* and *Enterococcaceae* were highly expressed in PD patients, while *Bifidobacteriaceae* and *Prevotellaceae* were increased in the rest of the patients.
Iguchi et al., 2020 [31]	Cohort study	38 (HD patients)	66.17 ± 12.38 (sucroferric oxyhydroxide group)	Baseline phyla in HD patients: *Firmicutes* 67.5%; *Proteobacteria* 11.0%; *Actinobacteria* 12.2%; *Bacteroides* 9.2%. PD patients had lower levels of *Bifidobacteriaceae* and *Prevotellaceae* compared to other groups, while *Enterobacteriaceae* and *Enterococcaceae* were increased.
Jiang et al., 2016 [32]	Case-control study	65 (CKD)	43.45 ± 16.90	Patients with advanced CKD and those with ESKD had significantly decreased abundance of *Roseburia* spp. and *F. prausnitzii* (respectively, *p* = 0.000 and *p* = 0.003).
		20 (healthy controls)	43.05 ± 9.88	
Jiang et al., 2017 [33]	Cross-sectional study	52 (ESKD patients)	51.58 ± 18.33	*E. coli*, *Bifidobacterium*, *Bacteroides fragilis* group, *Enterococcus* spp., *Clostridium coccoides* group, *Faecalibacterium prausnitzii*, *Roseburia* spp., and *Prevotella* were decreased in ESKD patients as compared to healthy controls. *Lactobacillus* group levels were similar in both groups.
		60 (healthy controls)	52.53 ± 13.98	
Kemp et al., 2021 [81]	Randomized, double-blind, controlled clinical trial	10 (Resistant starch type-2 group)	53.2 ± 12.3	*Firmicutes* phylum prevailed in HD patients. *Subdoligranum*, *Fusicatenibacter*, *Prevotella*, and *Blautia* were observed in HD patients.
		10 (placebo group)	55.1 ± 11.1	
Khiabani et al., 2022 [34]	Cross-sectional	20 (CKD/ESKD)	53.20 ± 12.03	*Clostridium* spp. abundance was similar in patients with CKD (including ESKD patients) and in healthy controls (*p* < 0.05).
		20 (healthy controls)	59.3 ± 7.89	
Kim et al., 2020 [35]	Cross-sectional study	103 (CKD stage 1–5)	48.9 ± 12.2 (ESKD)	*Alistipes*, *Oscillibacter*, *Lachnospira*, *Veillonella*, and *Dialister* were higher in control group as compared to patients with moderate to severe CKD. *Alistipes*, *Oscillibacter*, *Lachnospira*, and *Veillonella* were increased in mild CKD patients as compared to the moderate to severe CKD group.
		46 (healthy controls)	47.0 ± 10.8	
Kumar et al., 2021 [36]	Cross-sectional study	36 (IgAN)	45.5 ± 13.4	α-diversity was similar between IgAN patients and healthy controls but increased in advanced CKD stages (*p* = 0.025). *Fusobacteria* phylum was increased, while *Euryarchaoeota* phylum was decreased in patients with IgAN, as compared to healthy controls.
		12 (healthy controls)	46.5 ± 13.5	
Lai et al., 2019 [37]	Observational, prospective study	16 (CKD stages 3–4)	–	Patients with CKD had increased levels of *Bacteroidaceae*, *Enterobacteriaceae*, and *Rickenellaceae* as compared to healthy controls.
		16 (healthy controls)		
Lecamwasam et al., 2021 [38]	Observational, prospective study	95 (diabetes-associated CKD stages 1–5)	66.24 ± 10.22 (early CKD)	β-diversity and α-diversity were similar across all CKD stages. *Firmicutes* (the most abundant) and *Bacteroidetes* phyla abundance were similar in early and late CKD stages. *Prevotellaceae* was decreased across all CKD stages.
			72.68 ± 10.21 (late CKD)	
Li et al., 2019 [39]	Observational, prospective study	50 (CKD)	52.40 ± 13.49	The most prevalent bacteria in CKD patients were *Firmicutes* (42.27%), *Bacteroidetes* (37.85%), *Proteobacteria* (16.70%), *Actinobacteria* (1.48%), and *Verrucomicrobia* (0.67%). As compared to healthy controls, patients with CKD had decreased *Akkermansia* (*p* = 0.001) and *Parasutterella* (*p* = 0.007) levels, while *Lactobacillus* (*p* < 0.001), *Clostridium IV* (*p* = 0.015), and *Alloprevotella* (*p* < 0.001) levels were higher in CKD patients. *Akkermansia* associated with *Lactobacillus* had a good predictive value for CKD (AUC 0.830).
		22 (healthy controls)	50.27 ± 7.77	
Lin et al., 2021 [72]	Prospective, cohort study	109 (ESKD)	68.4 ± 10.4	In the high diversity group of patients, as well as in those with lower microbiota diversity, *Bacteroidetes*, *Firmicutes*, and *Proteobacteria* were the most prevalent phyla.
Lin et al., 2020 [41]	Case-control study	96 (HD)	68.1 ± 1.0	α-diversity was decreased in patients with normal-weight obesity as compared to those with normal weight or obesity (*p* = 0.001). *Firmicutes*/*Bacteroidetes* ratio was similar in all weight groups. *Faecalibacterium prausnitzii*, *Faecalibacterium*, and *Coprococcus* were decreased in the normal-weight obesity group.
Lin et al., 2020 [40]	Case-control study	88 (HD)	68.6 ± 11.0 (protein-energy wasting)	α-diversity was decreased in HD patients with moderate protein-energy wasting as compared to those with a normal nutritional status (*p* = 0.018). *Faecalibacterium prausnitzii* was significantly lower in protein-energy wasting patients, while *Akkermansia muciniphila* levels were higher.
			68.6 ± 9.7 (normal nutritional status)	
Lin et al., 2022 [42]	Case-control study	11 (ESKD)	30.93 ± 4.85	ESKD patients had a higher abundance of *Escherichia coli* (*p* < 0.001), *Bacteroides fragilis* (*p* = 0.010), *Bacteroides fragilis* (*p* = 0.010), and *Bacteroides caccae* (*p* = 0.047), as compared to healthy controls.
		11 (healthy controls)	27.99 ± 2.31	
Liu et al., 2021 [43]	Case-control study	100 (CKD)	56.64 ± 17.25	The Shannon index decrease was associated with CKD (*p* < 0.05). *Actinobacteria* levels were higher in CKD patients and predicted CKD prevalence (OR 1.037, 95% CI 1.007–1.068). Bacteroidetes was decreased in CKD patients and predicted CKD prevalence (OR 0.971, 95% CI, 0.951–0.991). *Bifidobacterium*, *Enterococcus*, and *Streptococcus* were more abundant in CKD patients.
		100 (healthy controls)	60.64 ± 16.51	
Liu et al., 2020 [83]	Randomized controlled trial	22 (HD + probiotic)	–	*Desulfovibrionaceae*, *Veillonellaceae*, and *Lactobacillaceae* abundance was increased in HD patients with diabetes mellitus, while *Halomonadaceae* and *Bradyrhizobiaceae* were decreased as compared to non-diabetic HD patients.
		23 (HD + placebo)		
Lun et al., 2018 [44]	Cross-sectional study	49 (CKD)	54 ± 14	Patients with CKD had increased abundance of *Bacteroidetes* and *Proteobacteria*, while *Firmicutes* was lower compared to healthy controls. *Ruminococcus gnavus* had the best discrimination power for CKD (AUC 0.764, 95% CI, 0.656–0.873, *p* = 0.000).
		24 (healthy controls)	56 ± 9	
Luo et al., 2021 [45]	Observational, cohort study	73 (ESKD)	49.71 ± 14.81 (HD)	HD and PD patients had an increased abundance of *Blautia* and *Dorea*, while *Prevotella* was decreased. *Akkermansia*, *Coprococcus*, *Acinetobacter*, *Proteus*, and *Pseudomonas* were increased in HD patients.
			48.95 ± 10.23 (healthy controls)	
Margiotta et al., 2020 [46]	Cross-sectional study	64 (CKD stages 3b-4)	80.7 ± 6.2	α-diversity was similar in CKD patients as compared to controls. Patients with CKD had a higher abundance of *Lactobacillus*, *Coprobacillus*, *Anaerotruncus*, *Citrobacter*, and *Ruminococcus torques*. Patients with CKD had lower levels of saccharolytic and butyrate-producing bacteria (*Prevotella* spp., *F. prausnitzii*, and *Roseburia* spp.).
		15 (healthy controls)	73.7 ± 7.6	
Al-Obaide et al., 2017 [47]	Cross-sectional study	20 (T2DM and advanced CKD)	64.4 ± 2.3	In patients with advanced CKD and T2DM, *Bifidobacterium* abundance was decreased, while *Clostridium*, *Escherichia*, *Enterobacter*, *Acinetobacter*, *Proteus*, and *Lactobacillus* levels were increased as compared to healthy controls.
		20 (healthy controls)	54.3 ± 3.2	
Pivari et al., 2022 [48]	Observational, cohort study	24 (CKD)	72 (67.5–78.8)	CKD patients had higher α-diversity as compared to healthy controls. CKD patients had decreased abundance of *Bacteroides* (*p* = 0.037), *Lachnoclostridium* spp. (*p* = 0.018), and *Escherichia*-*Shigella* (*p* = 0.048) compared to controls.
		20 (healthy controls)	74 (68.5–78.7)	
Ren et al., 2020 [49]	Observational, cohort study	110 (CKD)	51.75 ± 14.60	In CKD patients, 36 genera were increased (including *Klebsiella*, *Veillonella*, and *Desulfovibrio*), while 16 genera were decreased (including *Blautia*, *Roseburia*, and *Lachnospira*). *Clostridia*, *Verrucomicrobia*, and *Cyanobacteria* were decreased in CKD patients compared to controls.
		210 (healthy controls)	50.02 ± 4.56	
Salguero et al., 2019 [50]	Cross-sectional study	20 (T2DM and CKD)	62.8 ± 3.6	*Proteobacteria*, *Verrucomicrobia*, and *Fusobacteria* abundance were increased in CKD patients with T2DM as compared to healthy controls (*p* < 0.05 for all).
		20 (healthy controls)	58.5 ± 4.1	
Sato et al., 2021 [51]	Cross-sectional study	30 (early CKD)	68.83 ± 10.14	CKD patients had increased abundance of *Bacteroides coprocora* and *Bacteroides caccae*, while *Roseburia inulinivorans, Ruminococcus torques*, and *Ruminococcus lactaris* were more abundant in the non-CKD group.
		60 (non-CKD)	67.80 ± 11.48	
Simeoni et al., 2019 [88]	Randomized, placebo-controlled study	14 (CKD stage 3a + probiotics)	61.3 ± 5.2	*Lactobacillales* and *Bifidobacteria* had decreased levels in patients with CKD stage 3a (respectively, 2.3 × 10^3^ CFU/gr and 1.7 × 10^4^ CFU/gr).
		14 (CKD stage 3a, placebo)	58.2 ± 6.2	
Stadlbauer et al., 2017 [52]	Cross-sectional study	30 (dialysis patients)	61 (HD patients)	HD and PD patients had lower α-diversity index as compared to the control group (*p* < 0.05), but it was similar between HD and PD patients. *Blautia obeum*, *Clostridium citroniae*, and *Clostridium bolteae* levels were higher in HD patients compared to the control group. *Clostridium citroniae* and *Clostridium bolteae* were increased in PD patients compared to the control group. *Faecalibacterium prausnizii*, *Roseburia intestinalis*, and *Clostridium nexile* were decreased in HD patients compared to the control group.
		21 (healthy controls)	58	
Wang et al., 2019 [53]	Cross-sectional study	56 (CKD stages 1–4)	47.45 ± 15.47	Patients with CKD stage 5 had lower *Enterobacter*, *Enterococcus*, *Bifidobacterium*, *Bacteroides*, and *Clostridium* levels as compared to controls and patients with CKD stages 1–4 (*p* < 0.01 for all). *Faecalibacterium* and *Roseburia* were reduced in CKD stage 5 patients compared to CKD stages 1–4 and healthy controls (respectively, *p* = 0.018 and *p* < 0.001).
		72 (CKD stage 5)	51.69 ± 14.05	
		61 (healthy controls)	46.80 ± 10.47	
Wang et al., 2019 [54]	Observational, cohort study	28 (ESKD group)	43.9 ± 13.8	α-diversity was similar between ESKD and healthy controls group. Patients with ESKD had increased levels of *Prevotella*, *Faecalibacterium*, and *Fusobacterium*, while *Roseburia*, *Lachnospira*, *Dialister*, and *Bifidobacterium* abundance were decreased.
		19 (healthy controls)	44.1 ± 10.0	
Wang et al., 2020 [55]	Cross-sectional study	223 (ESKD group)	–	ESKD patients displayed a higher abundance of *Eggerthella lenta*, *Flavonifractor* spp., *Alistipes* spp., *Ruminococcus* spp., and *Fusobacterium* spp., while *Prevotella* spp., *Clostridium* spp., *Roseburia* spp., *Faecalibacterium prausnitzii*, and *Eubacterium rectale* were decreased.
		69 (healthy controls)		
Wu et al., 2020 [56]	Cross-sectional study	72 (CKD group)	65.00 ± 5.94 (advanced CKD)	*Bacteroides eggerthii* had a good discriminatory power between early-stage CKD and healthy controls (AUC = 0.80, 95% CI, 0.67–0.93), which was higher than in the case of protein/creatinine ratio (AUC = 0.64) and serum urea (AUC = 0.72).
		20 (non-CKD group)	64.00 ± 7.06	
Wu et al., 2020 [57]	Cross-sectional study	92 (CKD group)	66.2 ± 7.4 (advanced CKD)	CKD patients had increased abundance of *Bacteroides*, *Blautia*, *Escherichia-Shigella*, *Collinsella*, *Lachnoclostridium*, and *Lactobacillus*. *Paraprevotella* displayed a good discriminatory power between CKD patients and the control group (AUC = 0.78, 95% CI, 0.70–0.87).
		30 (control group)	61.6 ± 8.7	
Wu et al., 2021 [58]	Cross-sectional study	39 (CKD stages 4–5)	56.52 ± 15.72	CKD patients had increased abundance of *Proteobacteria*, *Enterobacteriaceae*, *Enterobacteriales*, *Gammaproteobacteria*, *Lactobacillales*, *Escherichia_Shigella*, *Enterococcus*, *Enterococcaceae*, and *Lactobacillaceae*.
		40 (healthy controls)	56.35 ± 10.96	
Xu et al., 2017 [59]	Cross-sectional study	32 (CKD group)	53.34 ± 14.47	*Enterobacteriaceae* and *Corynebacteriaceae* were more abundant in CKD patients as compared to controls, while *Ruminococcaceae* levels were decreased. *Enterococcus* and *Clostridium* were increased in CKD patients, whereas *Roseburia* and *Coprococcus* were decreased.
		32 (healthy controls)	55.03 ± 10.38	
Zhang et al., 2021 [60]	Cross-sectional study	46 (ESKD group)	Stratified in groups	*Ruminococcus gnavus*, *Ruminococcus* spp., *Eubacterium dolichum*, *Bacteroides ovatus*, and *Phascolarctobacterium* were more abundant in ESKD patients (including immunoglobulin A nephropathy), compared to healthy controls, while *Megamonas* spp., *Roseburia* spp., and *Eubacterium biforme* were decreased.
		15 (healthy controls)		
Zhang et al., 2020 [61]	Cross-sectional study	80 (CKD stages 3–5)	49.50 ± 24.80	*Megamonas*, *Megasphaera*, *Akkermansia*, *Lachnospira*, *Roseburia*, and *Fusobacterium* were increased in healthy controls as compared to patients with CKD and nephrotic syndrome. Patients with CKD and nephrotic syndrome had increased levels of *Parabacteroides*. *Oscillospira* and *Ruminococcus* were more abundant in the CKD group.
		48 (nephrotic syndrome)	48.47 ± 20.47	
		30 (healthy controls)	46.50 ± 22.67	
Zheng et al., 2020 [62]	Observational, cohort study	28 (ESKD)	43.9 ± 13.8	Patients with ESKD had increased levels of *Holdemania*, *Eggerthella*, and *Phascolarctobacterium*, while *Roseburia*, *Bifidobacterium*, and *Lachnospira* were decreased as compared to healthy controls.
		19 (healthy controls)	44.1 ± 10.0	

AUC = area under the curve; CKD = chronic kidney disease; ESKD = end-stage kidney disease; HD = hemodialysis; IgAN = immunoglobulin A nephropathy; PD = peritoneal dialysis; RRT = renal replacement therapy; T2DM = type 2 diabetes mellitus.

**Table 2 jcm-12-01948-t002:** Outcomes in CKD patients related to gut microbiota.

Study, Year	Outcomes	Results
Barros et al., 2015 [21]	Inflammation	VCAM-1 levels were negatively correlated with number of bands in CKD patients (r = −0.50, *p* = 0.03)
Ebrahim et al., 2022 [80]	Renal function decline	Creatinine levels were similar between intervention (β-glucan prebiotic) and control group during follow-up (14 weeks).
Jiang et al., 2017 [33]	Inflammation	*Roseburia* spp., *Faecalibacterium prausnitzii*, and *Prevotella* were negatively correlated with CRP (respectively, r = −0.452, *p* = 0.001; r = −0.431, *p* = 0.002 and r = −0.480, *p* = 0.000)
	Renal function	*Roseburia* spp., *Faecalibacterium prausnitzii*, *Clostridium coccoides* group, *Prevotella* were negatively correlated with Cystatin C levels (respectively, r = −0.414, *p* = 0.003; r = −0.395, *p* = 0.005; r = −0.400, *p* = 0.001 and r = −0.441, *p* = 0.001) *Bifidobacterium* was correlated with creatinine and blood urea nitrogen (r = −0.538, *p* = 0.000 and r = −0.495, *p* = 0.000, respectively)
Jiang et al., 2016 [32]	Inflammation	In CKD patients, *Roseburia* spp. and *F. prausnitzii* were negatively correlated with CRP (respectively, (r = −0.493, *p* = 0.00; r = -0.528, *p* = 0.000).
	Disease progression	In CKD patients, *Roseburia* spp. and *F. prausnitzii* were negatively correlated with Cystatin C (r = −0.321, *p* = 0.006; r = −0.445, *p* = 0.000) and positively corelated with eGFR (respectively, r = 0.347, *p* = 0.002 and r = 0.416, *p* = 0.000).
Lin et al., 2021 [72]	Mortality	The Simpson index and the Shannon index were lower in non-survivors as compared to patients who survived (respectively, *p* = 0.007 and *p* = 0.028). Non-survivors had higher levels of *Oscillospira*, *Achromobacter*, *Agrobacterium*, *PSB_M_3*, *Lactobacillus*, *vadinCA02*, *Alloscardovia*, *Anoxybacillus*, *Devosia*, and *Yersinia*.
Lin et al., 2020 [41]	Inflammation	The Shannon diversity index was negatively corelated with IL-6 (r = −0.253, *p* = 0.015) and TNFα (r = −0.260, *p* = 0.011). *Faecalibacterium prausnitzii* was negatively correlated with TNFα (r = −0.204, *p* = 0.047).
Lin et al., 2020 [40]	Inflammation	The Shannon diversity index was negatively corelated with IL-6 (r = −0.339, *p* = 0.001) and TNFα (r = −0.331, *p* = 0.002).
Luo et al., 2021 [45]	Mortality	ESKD patients with cardiovascular mortality had a lower proportion of *Bacteroides* and *Phascolarctobacterium* compared to survivors (*p* < 0.05).
	Peritonitis	PD patients with peritonitis had decreased *Dorea*, *Clostridium*, and *SMB53* proportions as compared to those without peritonitis (*p* < 0.05).
Margiotta et al., 2020 [46]	Inflammation	*Mogibacteriaceae* and *Oscillospira* were correlated with CRP levels. *Akkermansia*, *Ruminococcus*, and *Eubacterium* were negatively correlated with the neutrophil-to-lymphocyte ratio.
Zhou et al., 2022 [70]	Peritonitis	PD patients with *Escherichia coli* peritonitis had higher abundance of *Bacteroidetes* and *Synergistetes* compared to the non-peritonitis group, while *Bacilli* and *Lactobacillus* were decreased.
Zhu et al., 2022 [69]	Responsiveness to erythropoietin	*Neisseria*, *Streptococcus*, *Porphyromonas*, *Fusobacterium*, *Prevotella_7*, *Rothia*, *Leptotrichia*, *Prevotella*, and *Actinomyces* could predict a poor response to erythropoietin in ESKD patients. *Neisseria* had an excellent power to discriminate between good and poor response to erythropoietin in ESKD patients (AUC 0.9535, 95% CI, 0.902–1.0, *p* < 0.0001).

CKD = chronic kidney disease; CRP = C-reactive protein; ESKD = end-stage kidney disease; IL-6 = interleukin 6; PD = peritoneal dialysis; TNFα = tumor necrosis factor alpha; VCAM-1 = vascular cell adhesion molecule 1.

**Table 3 jcm-12-01948-t003:** Studies reporting gut microbiota modulation in CKD patients.

Study, Year	Type of Therapy	Results
Abdelbary et al., 2022 [75]	Sucroferric oxyhydroxide	In hemodialysis patients, *Veillonella* spp. and *Ruminococcus torques* levels increased (*p* = 0.0351 for both), while *Subdoligranulum* decreased (*p* = 0.0496).
Belova et al., 2020 [78]	Immobilized synbiotic LB-complex L vs. placebo	In 56% of patients in the treatment group, gut microbiota recovered as compared to placebo (grade III dysbiosis was absent after therapy). CRP decreased from 6.8 ± 3.1 g/L to 5.3 g/L in the treatment group.
Borges et al., 2017 [79]	Probiotics	Gut microbiota profile was similar in the probiotic group (*Streptococcus thermophilus*, *Lactobacillus acidophilus*, and Bifidobacteria longum strains) and placebo group after 3 months of therapy (similar number of bands).
Ebrahim et al., 2022 [80]	β-glucan prebiotic	*Prevotella* tended to increase in the intervention group (β-glucan) as compared to the control group, while *Bacteroides* and *Blautia* tended to decrease.
Hu et al., 2022 [66]	Dietary intervention	HD patients from the protein-energy wasting group had lower abundance of *Roseburia* as compared to HD patients in the non-protein energy wasting group (*p* = 0.022). *Escherichia* abundance was increased in PD patients from the protein-energy wasting group compared to PD patients from the non-protein-energy wasting group (*p* = 0.022).
Iguchi et al., 2020 [31]	Sucroferric oxyhydroxide	In HD patients, sucroferric oxyhydroxide did not affect major phyla (*p* = 0.849 for *Firmicutes*, *p* = 0.776 for *Proteobacteria*, *p* = 0.517 for *Actinobacteria*, *p* = 0.728 for *Bacteroides*).
Jiang et al., 2020 [67]	Dietary intervention	Patients with CKD stage 5 who received a very low protein diet had higher levels of *Escherichia*, *Shigella*, and *Klebsiella*, while *Blautia* was decreased.
Kemp et al., 2021 [81]	Resistant starch type-2	Resistant starch type-2 increased *Oscillosperaceae*, *Roseburia*, and *Ruminococcus gauvreauii* levels. Resistant starch type-2 decreased *Ruminococcus champanellens*, *Dialister*, and *Coprococcus*.
Kimber et al., 2020 [82]	Rifaximin	Rifaximin was linked to reduced diversity and richness of microbiota as compared to placebo. Rifaximin reduced 10 bacterial taxa from *Firmicutes* and *Actinobacteria* phyla (including *Clostridium*, *Turicibacter*, and *Anaerotruncus*).
Laffin et al., 2019 [89]	Amylose-resistant starch	Amylose-resistant starch increased levels of *Faecalibacterium* in ESKD patients as compared to placebo (from 0.40 ± 0.50% to 3.21 ± 4.97%, *p* = 0.03), while *Parabacteroides*, *Bifidobacteria*, *Ruminococcus*, and *Prevotella* levels did not change.
Lai et al., 2019 [37]	Low-protein diet	Low-protein diet increased *Akkermansiaceae* and *Bacteroidaceae* and decreased *Christensenellaceae*, *Clostridiaceae*, *Lactobacillaceae*, and *Pasteurellaceae* levels.
	Low-protein diet + inulin	Low-protein diet associated with inulin therapy increased *Bifidobacteriaceae* levels.
	Inulin	Inulin decreased *Enterobacteriaceae* family.
Liu et al., 2020 [83]	Probiotics	Probiotics increased *Bacteroidaceae* and *Enterococcaceae* abundance compared to placebo. Probiotics decreased *Ruminococcaceae*, *Halomonadaceae*, *Peptostreptococcaceae*, and *Clostridiales Family XIII* levels compared to placebo.
Liu et al., 2022 [76]	Iron supplementation	After oral iron supplementation, α-diversity and *Firmicutes* levels decreased, while *Bacteroides* increased. Moreover, *Blautia* and *Coprococcus* levels decreased, while *Bacteroidetes* increased.
McFarlane et al., 2021 [84]	Synbiotics vs. placebo	Synbiotic therapy increased *Bifidobacterium animalis* (*p* < 0.001) and *Blautia* spp. levels (*p* = 0.004). Synbiotics decreased *Bacteroides cellulosilyticus* and *Ruminiclostridium* spp. (*p* < 0.05 for both). Synbiotic therapy was linked to eGFR decrease with 3.14 mL/min/1.73 m^2^ (*p* < 0.01).
Miao et al., 2018 [63]	Lanthanum carbonate	In HD patients, lanthanum carbonate decreased *Bacteroides* and *Proteobacteria* but increased *Actinobacteria* levels. Shannon index decreased following lanthanum carbonate therapy.
Cruz-Mora et al., 2014 [85]	Synbiotics	In HD patients, synbiotic therapy increased *Bifidobacterium* abundance (*p* = 0.0344) but decreased *Lactobacillus* levels.
Nazzal et al., 2017 [77]	Oral vancomycin	Following vancomycin therapy, *Clostridia*, *Roseburia*, *Enterococcaceae*, and *Bacteroidales* decreased, while *Veillonellaceae* increased.
Pivari et al., 2022 [48]	Curcumin supplementation	After 6 months of dietary intervention, *Escherichia*-*Shigella* levels significantly decreased, while *Lachnoclostridium* and *Lactobacillaceae* spp. increased.
Rocchetti et al., 2021 [86]	Dietary intervention	The keto analogs-supplemented Mediterranean diet reduced *Clostridiaceae*, *Methanobacteriaceae*, *Prevotellaceae*, and *Lactobacillaceae* abundance, while *Bacteroidaceae* and *Lachnospiraceae* levels increased.
Rossi et al., 2016 [87]	Synbiotics	Compared to placebo, synbiotics were linked to a 5-fold increase in *Bifidobacterium* spp. (*p* = 0.003), while *Lactobacillus* spp. abundance was similar.
Simeoni et al., 2019 [88]	Probiotics	Compared to the placebo group, probiotics increased *Lactobacillales* and *Bifidobacteria* levels from 2.1 × 10^3^ CFU/gr and 1.9 × 10^4^ CFU/gr to 2.2 × 10^6^ CFU/gr and 2.5 × 10^7^ CFU/gr, respectively (*p* < 0.001 for both). Iron and ferritin levels were significantly increased after probiotic therapy (*p* < 0.001 for both), while CRP, total cholesterol, and triglycerides levels were decreased in patients who received probiotics (respectively, *p* < 0.001, *p* < 0.01, and *p* < 0.01).
Wu et al., 2020 [64]	Dietary intervention	CKD patients who received a low protein diet had lower levels of *Lachnospiraceae* and *Bacteroidaceae* as compared to those receiving a normal protein diet.
Wu et al., 2020 [65]	Phosphate binders	α-diversity and Simpson index were decreased in HD patients receiving calcium carbonate compared to the ferric citrate group (respectively, *p* = 0.049 and *p* = 0.001). Patients receiving ferric citrate had increased levels of *Bacteroidetes* phylum levels, while *Firmicutes* phylum was decreased.
Yacoub et al., 2017 [68]	Advanced glycation end products	PD patients who received a one-month advanced glycation end-product restriction had a lower abundance of *Prevotella copri* compared to those with a normal diet.

CKD = chronic kidney disease; CRP = C-reactive protein; eGFR = estimated glomerular filtration rate; HD = hemodialysis; PD = peritoneal dialysis.

## Data Availability

The data presented in this study are available on request from the corresponding author.

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
