# Peer review of "Gut Microbiota in Chronic Kidney Disease: From Composition to Modulation towards Better Outcomes—A Systematic Review"

_jcm, 2023, doi:10.3390/jcm12051948_

Round 1

Reviewer 1 Report

1. Please delete some keywords. I think some of them cannot be keywords.

2. Please check the manuscript carefully, I found some spell mistakes. Such as Line 108( thirst--first?), Hanifi et al, 2021 in table 1(ESDK--ESKD?), Line 227 (Figure 3--figure 2?).

3. Generally speaking, it is a general systematic review with certain scientific significance, but there is no detailed literature comparison and discussion. Some better reviews have been published recently, especially one paper in Nature Review Nephrology(Gut microbiome studies in CKD: opportunities, pitfalls and therapeutic potential).

Reviewer 2 Report

This is an systematic review of gut microbiota in CKD analyzing microbiota composition and the modulation effect of some strategies, but not on outcomes as the authors claim in the title. In fact, the authors mention the modulatory effects of some interventions on changes in gut microbiota, which may be interesting, but it could be more relevant to evaluate changes in inflammation, uremic toxins or CKD progression, the consequences of the gut disbiosis in CKD. (e.g. Zhu H, Cao C, Wu Z, Zhang H, Sun Z, Wang M, Xu H, Zhao Z, Wang Y, Pei G, Yang Q, Zhu F, Yang J, Deng X, Hong Y, Li Y, Sun J, Zhu F, Shi M, Qian K, Ye T, Zuo X, Zhao F, Guo J, Xu G, Yao Y, Zeng R. The probiotic L. casei Zhang slows the progression of acute and chronic kidney disease. Cell Metab. 2021 Oct 5;33(10):1926-1942). This review is too ambitious since there is a lot of literature to review and summarize in a single manuscript. Further, the manuscript do no add further knowlegde to the field with respect to previous systematic reviews that the authors mention. In addition, there are many flaws in the study that precludes its publication. e.g there are references missing which questions the validity of the search. In the results there are also many flaws

Introduction:

In some parts references supporting the sentence (e.g the bidirectional relationship between gut and kidney, or lines 66, 72, 73). Is colonic transit (vs transition) time (line 59)

Methods

Why did the authors in the search strategy did not include “peritoneal dialysis” or “kidney trasplantation” ?

Results:

There are some flaws and I miss some references in the results. As reviewer I do not have time to review all the data and do an extensive search as the authors did, but I found some important deficiencies, and likely there are more

Table 1 In some studies reported, specially those from RCT there is not a healthy control group, eg Belova 2022, Kemp 2021 making the interpretation of the results difficult/questionable. in the reference Ebrahim 2022 et al the authors do not compare CKD patients with healthy controls, only describe the baseline findings and compare CKD patients of South Africa with the Flemish Gut Flora Project as a background (different populations). In the case of He 2021 the authors claim in the abstract that there is a control group but only mention CKD patients not on dialysis and on dialysis. In the webpage of the journal Niger J Clin Pract (quality ?) volume 24 is not available. He 2021 is not included in the references. There are 3 Hu in the table but I only find 2 references (24,25) Ref 24 describes discrimination but not differences in expression of bacteria genera which is more relevant. Ref 25 is not well described in the table 166 CKD patients (including CKD, HD and PD) and 17 controls and interesting data are missing. The ref lacking is likely Hu X, Ouyang S, Xie Y, Gong Z, Du J. Characterizing the gut microbiota in patients with chronic kidney disease. Postgrad Med. 2020 Aug;132(6):495-505. and compared 95 CKD non dialysis and 20 healthy controls which are lacking in the description of the study, The main findings increase in Proteobacteria and decrease in Sinergistestes and butirate-producing bacteria is not mentioned.

Ref Lin TY, Wu PH, Lin YT, Hung SC. Gut dysbiosis and mortality in hemodialysis patients. NPJ Biofilms Microbiomes. 2021 Mar 3;7(1):20. doi: 10.1038/s41522-021-00191-x.  refered in table 1 and 2 is lacking

Ref He H, Xie Y. Effect of Different Hemodialysis Methods on Microbiota in Uremic Patients. Biomed Res Int. 2020;2020:6739762. doi: 10.1155/2020/6739762. is not included in the review

Table 2 does report only outcomes but associations since include cross sectional studies. Further, the study of Ebrahim (no changes in serum creatine between groups in only 14 weeks of follow up is not clinically relevant).  The relationship between some species of gut microbiota and inflammation reported in ref 24 is lacking. Jiang 2016 shows association between Roseburia spp and F. prausnitzii an d eGFR, thus no association with CKD progression

Effects of drugs/interventions on gut microbiota (table 3). Some articles are missing

-Laffin MR, Tayebi Khosroshahi H, Park H, Laffin LJ, Madsen K, Kafil HS, Abedi B, Shiralizadeh S, Vaziri ND. Amylose resistant starch (HAM-RS2) supplementation increases the proportion of Faecalibacterium bacteria in end-stage renal disease patients: Microbial analysis from a randomized placebo-controlled trial. Hemodial Int. 2019 Jul;23(3):343-347. doi: 10.1111/hdi.12753.

-Raj DS, Sohn MB, Charytan DM, Himmelfarb J, Ikizler TA, Mehrotra R, Ramezani A, Regunathan-Shenk R, Hsu JY, Landis JR, Li H, Kimmel PL, Kliger AS, Dember LM; Hemodialysis Novel Therapies Consortium. The Microbiome and p-Inulin in Hemodialysis: A Feasibility Study. Kidney360. 2021 Jan 15;2(3):445-455. doi: 10.34067/KID.0006132020. 

-Pivari F, Mingione A, Piazzini G, Ceccarani C, Ottaviano E, Brasacchio C, Dei Cas M, Vischi M, Cozzolino MG, Fogagnolo P, Riva A, Petrangolini G, Barrea L, Di Renzo L, Borghi E, Signorelli P, Paroni R, Soldati L. Curcumin Supplementation (Meriva®) Modulates Inflammation, Lipid Peroxidation and Gut Microbiota Composition in Chronic Kidney Disease. Nutrients. 2022 Jan 5;14(1):231. doi: 10.3390/nu14010231. 

Mentioned previously in table 1 with a comparison between CKD and healthy controls but the results of the intervention are not mentioned in table 3

-Abdelbary MMH, Kuppe C, Michael SS, Krüger T, Floege J, Conrads G. Impact of sucroferric oxyhydroxide on the oral and intestinal microbiome in hemodialysis patients. Sci Rep. 2022 Jun 10;12(1):9614. doi: 10.1038/s41598-022-13552-z.

-Liu H, Wu W, Luo Y. Oral and intravenous iron treatment alter the gut microbiome differentially in dialysis patients. Int Urol Nephrol. 2022 Sep 27. doi: 10.1007/s11255-022-03377-0.

-Nazzal L, Roberts J, Singh P, Jhawar S, Matalon A, Gao Z, Holzman R, Liebes L, Blaser MJ, Lowenstein J. Microbiome perturbation by oral vancomycin reduces plasma concentration of two gut-derived uremic solutes, indoxyl sulfate and p-cresyl sulfate, in end-stage renal disease. Nephrol Dial Transplant. 2017 Nov 1;32(11):1809-1817.

Further, the explanation of the results by CKD stage or type of dialysis explain the results of some studies, rather than summarizing the results of the different studies

Discussion

Generally is a brief description of the results and with limited interpretation of them. Further, some results deserve to be relativized (eg Study of Luo 2021 with 5 deaths and 7 peritonitis the differences between groups should be relativized)

The efffects of pre, pro and simbiotics is summarized in the results of a previous metaanalysis and the effects of other interventions (oral iron, phosphate binders( is not discussed

References

Some references are incomplete and lack pages e.g ref 7

The article requires english review and correction of misspellings

Reviewer 3 Report

This is an interesting review. Although I am not familiar with that topic I found that not all the issues were included. Furthermore, the authors present a list of studies without attempting an analytical approach in the form of a metaanalysis, such as plots and presebtation on of HR or OR. This is more suprising because they used bias assessment tools typical for metaanalyses, There also many other methodological drawbacks.

Lacki of definitions regarding the terms prebiotics probiotics and symbiotics

The work of each author should be stated in the methods section

How was early CKD defined ?

The results should be more carefully presented. For example, in the title early CKD, the paragraph beginns with a comparison of early CKD with controls, then early CKD and CKD are used interchangeably.

The authors should decide how to present their results. The listing in a table of all published studies does not help, It is better to synthethize the data in the context of a metanalysis or present only the most relevant of them in a table wihtout describing microbiologic data difficult to recapitulate. This should be made in the text.

Are there any data on patients with peritoneal dialysis, transplantation, or patients with glomerulonephritis ?This should be explained in the discussion or mentioned in the methods if such studies were excluded.

Were there any data regarding intradialytic hypotension , which is a risk factor for intestinal iscemia

Round 2

Reviewer 3 Report

The authors addressed sufficiently all my comments.

I have no other objections to the publication of the revised version of the paper.